# Synthesis and Single Crystal Structures of *N*-Substituted Benzamides and Their Chemoselective Selenation/Reduction Derivatives

**DOI:** 10.3390/molecules26082367

**Published:** 2021-04-19

**Authors:** Guoxiong Hua, Cameron L. Carpenter-Warren, David B. Cordes, Alexandra M. Z. Slawin, J. Derek Woollins

**Affiliations:** 1EaStCHEM School of Chemistry, University of St Andrews, St Andrews Fife KY16 9ST, UK; gh15@st-andrews.ac.uk (G.H.); clcw@st-andrews.ac.uk (C.L.C.-W.); dbc21@st-andrews.ac.uk (D.B.C.); amzs@st-andrews.ac.uk (A.M.Z.S.); 2Department of Chemistry, Khalifa University, Abu Dhabi 127788, United Arab Emirates

**Keywords:** *N*-Substituted Benzamides, Woollins’ reagent, selenation reagent, reduction reagent, single crystal X-ray structures

## Abstract

A series of *N*-aryl-*N*-(2-oxo-2-arylethyl) benzamides and cinnamides has been prepared. The reaction of the benzamides with Woollins’ reagent, a highly efficient chemoselective selenation/reduction reagent, gave the corresponding *N*-aryl-*N*-(arylenethyl) benzoselenoamides in good yields. Five representative single crystal X-ray structures are discussed.

## 1. Introduction

2,4-Bis (phenyl)-1,3-diselenadiphosphetane-2,4-diselenide (Woollins’ reagent, WR) has played a role in synthetic chemistry in the past two decades [1,2,3,4,5,6,7,8,9,10,11,12,13]. It has been successfully applied as an efficient building unit to synthesize a series of eight-, nine-, and ten-membered selenophosphorus heterocycles with P-Se-Se-P linkage [14], as well as unique octaselenocyclododecane with four carbon atoms and eight selenium atoms in this twelve-membered cycle [15]. Another attractive application has been that it acts as a highly chemoselective reagent, e.g., the reduction of a wide range of 1,4-enediones and 1,4-ynediones in methanol led to saturated 1,4-diketones [16] and the selective reduction of the double bond of 2-α,β-unsaturated thiazo- and selenazolidinones gave the corresponding saturated heterocycles [17]. Woollins’ reagent has also been used as a reducting agent to transfer porpholactone into dihydroporpholactone or into adjacent-tetrahydroporpholactone [18].

Organoselenium compounds have received growing attention during the last decades due to their importance as useful precursors in synthetic chemistry [19,20], as new synthetic materials [20], and their biological and medicinal significance [21]. To continue our interest in the chemistry of Woollins’ reagent towards various organic substrates, we report an investigation on the use of WR as a selenation/reduction reagent for transferring *N*-aryl-*N*-(2-oxo-2-arylethyl) benzamides into the corresponding *N*-aryl-*N*-(arylenethyl) benzoselenoamides.

## 2. Results and Discussion

### 2.1. Synthesis and Characterization

The synthesis of anilinoacetophenones **1**–**3**, *N*-aryl-*N*-arylamidoacetophenones **4**–**6** and *N*-aryl-*N*-cinnamidoacetophenones **7**–**9** was carried out using a modified literature method [22]. The reaction of anilines and an equivalent of the appropriate bromoacetophenones in dry acetonitrile at room temperature gave anilinoacetophenones **1**–**3** in 81–87% yields, respectively. Anilinoacetophenones **1** and **2** are new compounds, while **3** is a known compound, prepared previously by a similar method [23,24]; however, its single-crystal X-ray structure has not been reported previously. Acylation of anilinoacetophenones **1**–**3** with the appropriate acid chlorides in 1,2-dichloroethane at reflux led to the *N*-aryl-*N*-arylamidoacetophenones **4**–**6** and *N*-aryl-*N*-cinnamidoacetophenones **7**–**9** in 76–91% yields, as shown in Scheme 1. All the new compounds were characterized by standard analytical and spectroscopic techniques. **1**–**9** show the anticipated [M]^+^ or [M + H]^+^ peak in their mass spectra, satisfactory accurate mass measurements, and appropriate isotopic distributions; the ^1^H NMR spectra display all the characteristic peaks of the phenyl backbones in compounds and the characteristic peaks of the NH group in compound **1**–**3**. The ^13^C NMR spectra of compounds **1**–**9** display the characteristic signals of the C=O groups.

Selenation of *N*-aryl-*N*-arylamidoacetophenones **4**–**6** by WR gave rise to *N*-aryl-*N*-arylethylbenzoselenoamides **10**–**12** in 50%, 46% and 40% yields, respectively, rather than the expected 1,3-selenazole products (Scheme 2). One C=O group has been converted to C=Se and the other reduced to CH_2_ to give the final product *N*-Aryl-*N*-arylethylbenzoselenoamides **10**–**12**. It is well known that WR is an efficient chemoselective reduction agent for diketones [16], α,β-unsaturated thioazo and selenoazolidinones [6]. Based on the literature research and our findings, a possible mechanism for the selective reduction of *N*-substituted-*N*-phenylamidoacetophenones **4**–**6** is broadly similar to that of NaSeH and LiSeH as selective reducing agents of α,β-unsaturated carbonyl compounds [13] and of PhSe-SePh as a reducing agent for electron deficient olefins [25], and it is probable that the reduction proceed through a Micheal reaction [26,27,28].

Reacting *N*-aryl-*N*-cinnamidoacetophenones **7**–**9** with WR under similar reaction conditions did not lead to a reaction, with the starting materials being recovered (Scheme 3). We speculate that the extra C=C bond in *N*-substituted-*N*-phenylamidoacetophenones **7**–**9** which gives a conjugated structure, may be more stable and robust than *N*-substituted-*N*-phenylamidoacetophenones **4**–**6** towards WR.

The new selenium derivatives **10**–**12** are quite stable both as solids and in solution, in air and moist atmospheres, and are soluble in common organic solvents. They show the anticipated molecular ion peaks [M + H]^+^ in their CI spectra and [M]^+^ in their EI spectra, and satisfactory accurate mass measurements (EI). All the characteristic peaks of the phenyl backbones were found, and the characteristic peaks of the NH group disappeared in their ^1^H NMR spectra. Their ^13^C NMR spectra all show the normal signals for the C=Se groups (δ_C_, 204.8–206.8 ppm). Their ^77^Se NMR spectra exhibit singlet signals at δ_Se_ = 598.4, 601.5 and 601.4 ppm for **10**–**12**, respectively.

### 2.2. Single Crystal Structure Analysis

Single crystals of **3**, **7**, **10**–**12** suitable for X-ray crystallographic analysis were grown by slow evaporation of dichloromethane solutions of the compound in air at room temperature. Selected crystallographic data are given in Table 1 and the resulting molecular structures are illustrated in Figure 1 and Figure 2.

The molecular structure of anilinoacetophenone **3** (Figure 1) shows a planar arrangement, with a mean deviation of non-hydrogen atoms from the plane of 0.047 Å. Adjacent molecules of **3** interact to form hydrogen-bonded dimers via a pair of NH···O hydrogen bonds at a H···O distance of 2.59(4) Å, and N···O separation of 3.360(5) Å. The structure of the *N*-aryl-*N*-cinnamidoacetophenone **7** shows the compound in a *twisted-T* conformation (Figure 1). As expected, the acetophenone group retains its planarity (mean deviation of non-hydrogen atoms from the plane of 0.004 Å). Meanwhile, the other two phenyl ring planes are twisted out of the acetophenone plane, with angles between planes of 71.36 and 50.74° for C(10)-C(15) and C(21)-C(26), respectively.

The X-ray structures of **10**, **11** and **12** are depicted in Figure 2, each displaying similar *twisted-T* conformations. The three aryl rings [C(5)-C(10) (ring 1), C(11)-C(16) (ring 2) and C(18)-C(23) (ring 3) in all three compounds] are all twisted with respect to each other, the angles between ring planes being 28.92, 69.02 and 62.00° in **10**, 18.92, 62.23 and 64.51° in **11** and 55.76, 61.91 and 68.23° in **12** for rings 1 and 2, 1 and 3, and 2 and 3, respectively. This pattern of ring twists puts all the rings out of the plane of the selenoamide, except for ring 1 in compound **12**, which is near parallel to the selenoamide, inclined at 9.24°. The C=Se double bond lengths [1.832(3) Å in **10**, 1.833(2) Å in **11** and 1.8264(19) Å in **12**] are very similar, falling at the middle of the range of C=Se bonds in known selenoamides [1.81(5) to 1.856(4) Å]. [29] The conformations of **10**–**12** are all very similar, that between **10** and **11**, being almost identical. The difference between this conformation and that adopted by **12** is in the orientation of ring 1, which differs by ~67° between the two conformations.

## 3. Materials and Methods

### 3.1. General

Unless otherwise stated, all reactions were carried out under an oxygen-free nitrogen atmosphere using pre-dried solvents and standard Schlenk techniques, subsequent chromatographic and work up procedures were performed in air. ^1^H (400.1 MHz), ^13^C (100.6 MHz) and ^77^Se-{^1^H} (51.5 MHz, referenced to external Me_2_Se) NMR spectra were recorded at 25 °C (NMR Jeol GSX270). IR spectra were recorded as KBr pellets in the range of 4000–250 cm^−1^ on a Perkin-Elmer 2000 FTIR/Raman spectrometer Mass spectrometry was performed by the EPSRC National Mass Spectrometry Service Centre, Swansea.

#### Crystallography

X-ray diffraction data for compounds **3**, **7** and **10**–**12** were collected at either 93 K or 173 K using a Rigaku FR-X Ultrahigh Brilliance Microfocus RA generator/confocal optics with XtaLAB P200 diffractometer [Mo Kα radiation (λ = 0.71075 Å)]. Intensity data were collected using ω steps accumulating area detector images spanning at least a hemisphere of reciprocal space. Data for all compounds were collected using CrystalClear 2.1 [30] and processed (including correction for Lorentz, polarization and absorption) using either CrystalClear [30] or CrysAlisPro 1.171.38.43. [31] Structures were solved by dual-space (SHELXT-2014/4 [32]) or Patterson (PATTY [33]) methods and refined by full-matrix least-squares against F^2^ (SHELXL-2018/3 [34]). Non-hydrogen atoms were refined anisotropically, and hydrogen atoms were refined using a riding model, except for the amine hydrogen in **3** which was located from the difference Fourier map and refined isotropically subject to a distance restraint. All calculations were performed using the CrystalStructure 4.3 interface [35]. Selected crystallographic data are presented in Table 1. Deposition numbers 2071413–2071417 contains the supplementary crystallographic data for this paper. These data are provided free of charge by the joint Cambridge Crystallographic Data Centre and Fachinformationszentrum Karlsruhe Access Structures service www.ccdc.cam.ac.uk/structures.

### 3.2. Synthesis

#### 3.2.1. General Procedure for Synthesis of Compounds 1–3

The appropriate aniline (20 mmol) and the phenacyl bromide (10 mmol) were combined in MeCN (40 mL) and allowed to stir at room temperature for 24 h. The amine salt was filtered off and the filtrate was concentrated under vacuum. The residue was dissolved in EtOAc (40 mL) and washed sequentially with H_2_O (50 mL), 5% citric acid (50 mL) and brine (25 mL). The organic layer was dried over Na_2_SO_4_, filtered through a pad of silica gel and the solvent was evaporated to give the product aminoacetophenones **1**–**3** in good yield.

1-(4-Bromophenyl)-2-((4-ethylphenyl)amino)ethan-1-one (**1**). Brown solid (83% yield). M.p. 151–153 °C. Selected IR (KBr, cm^−1^), 1679(vs), 1617(s), 1583(s), 1522(vs), 1585(m), 1388(m), 1351(m), 1311(m), 1216(m), 1178(m), 1140(m), 1068(s), 992(vs), 812(vs), 576(m), 501(m). ^1^H NMR (CDCl_3_, δ), 7.80 (d, *J*(H,H) = 8.6 Hz, 2H), 7.69 (d, *J*(H,H) = 8.4 Hz, 2H), 7.32 (s, 1H), 7.09 (d, *J*(H,H) = 8.6 Hz, 2H), 6.68 (d, *J*(H,H) = 8.6 Hz, 2H), 4.60 (s, 2H), 2.58 (d, *J*(H,H) = 7.6 Hz, 2H), 1.22 (d, *J*(H,H) = 7.6 Hz, 3H) ppm. ^13^C NMR (CDCl_3_, δ), 194.4, 145.0, 133.9, 133.7, 132.2, 129.3, 129.0, 128.7, 113.2, 50.7, 28.0, 16.0 ppm. Accurate mass measurement [ESI^+^, *m*/*z*]: found 318.0482 [M + H]^+^, calculated mass for C_16_H_16_BrNOH: 318.0486.

1-(4-Chlorophenyl)-2-((4-ethylphenyl)amino)ethan-1-one (**2**). Dark yellow solid (81% yield). M.p. 148–149 °C. Selected IR (KBr, cm^−1^), 1678(vs), 1618(m), 1598(s), 1522(s), 1489(m), 1440(m), 1395(m), 1352(s), 1312(m), 1218(s), 1090(s), 993(s), 815(vs), 577(m), 529(m). ^1^H NMR (CDCl_3_, δ), 7.99 (d, *J*(H,H) = 8.6 Hz, 2H), 7.52 (d, *J*(H,H) = 8.6 Hz, 2H), 7.32 (s, 1H), 7.09 (d, *J*(H,H) = 8.4 Hz, 2H), 6.68 (d, *J*(H,H) = 8.4 Hz, 2H), 4.41 (s, 2H), 2.60 (q, *J*(H,H) = 7.6 Hz, 2H), 1.23 (t, *J*(H,H) = 7.6 Hz, 3H) ppm. ^13^C NMR (CDCl_3_, δ), 194.2, 145.0, 140.3, 133.9, 133.3, 129.3, 129.2, 129.1, 128.7, 113.2, 50.8, 28.0, 16.0 ppm. Accurate mass measurement [ESI^+^, *m*/*z*]: found 274.0992 [M + H]^+^, calculated mass for C_16_H_16_ClNOH: 274.0994.

2-((4-Bromophenyl)amino)-1-(4-chlorophenyl)ethan-1-one (**3**). Greenish yellow solid (87% yield). M.p. 165–166 °C. Selected IR (KBr, cm^−1^), 1678(vs), 1595(s), 1510(s), 1491(s), 1400(m), 1357(s), 1256(m), 1218(m), 1094(s), 991(s), 814(s), 797(m), 574(m), 499(m). ^1^H NMR (CDCl_3_, δ), 7.98 (d, *J*(H,H) = 8.6 Hz, 2H), 7.52 (d, *J*(H,H) = 8.6 Hz, 2H), 7.32 (d, *J*(H,H) = 8.4 Hz, 2H), 7.28 (s, 1H), 6.60 (d, *J*(H,H) = 8.4 Hz, 2H), 4.56 (s, 2H) ppm. ^13^C NMR (CDCl_3_, δ), 194.2, 145.0, 140.3, 133.9, 133.3, 129.3, 129.2, 129.2, 128.7, 113.2, 50.8, 28.0, 16.0 ppm. Accurate mass measurement [ESI^+^, *m*/*z*]: found 323.9789 [M + H]^+^, calculated mass for C_14_H_11_BrClNOH: 323.9791.

#### 3.2.2. General Procedure for Synthesis of Compounds 4–9

The appropriate aminoacetophenone (5.0 mmol) was dissolved in dichloroethane (25 mL) and refluxed for 2 h with the appropriate acid chloride (5.0 equiv). Volatiles were evaporated in vacuo, and the residue was recrystallized from ethyl acetate to give the expected products **4**–**9**.

*N*-(2-(4-Bromophenyl)-2-oxoethyl)-*N*-(4-ethylphenyl)-4-methoxybenzamide (**4**). Yellow solid (91% yield). M.p. 127–129 °C. Selected IR (KBr, cm^−1^), 1678(vs), 1617(m), 1590(m), 1521(s), 1352(m), 1306(m), 1262(m), 1219(m), 1091(m), 994(s), 844(m), 815(s), 772(m), 696(m), 613(m), 546(m), 503(m). ^1^H NMR (CDCl_3_, δ), 8.09 (d, *J*(H,H) = 8.9 Hz, 2H), 7.90 (d, *J*(H,H) = 8.7 Hz, 2H), 7.68 (d, *J*(H,H) = 8.6 Hz, 2H), 7.09 (d, *J*(H,H) = 8.6 Hz, 2H), 6.97 (d, *J*(H,H) = 8.9 Hz, 2H), 6.68 (d, *J*(H,H) = 8.5 Hz, 2H), 4.59 (s, 2H), 3.91 (s, 3H), 2.59 (q, *J*(H,H) = 7.6 Hz, 2H), 1.22 (t, *J*(H,H) = 7.6 Hz, 3H) ppm. ^13^C NMR (CDCl_3_, δ), 194.5, 171.3, 164.2, 145.0, 133.9, 133.7, 132.4, 133.2, 132.0, 129.3, 129.0, 128.7, 121.7, 113.8, 113.3, 55.5, 50.8, 28.0, 16.0 ppm. Accurate mass measurement [CI^+^, *m*/*z*]: found 452.0859 [M + H]^+^, calculated mass for C_24_H_22_BrNO_3_H: 452.0861.

*N*-(2-(4-Chlorophenyl)-2-oxoethyl)-*N*-(4-ethylphenyl)-4-methoxybenzamide (**5**). Bright yellow solid (81% yield). M.p. 126–128 °C. Selected IR (KBr, cm^−1^), 1678(vs), 1617(s), 1590(s), 1512(vs), 1396(m), 1352(m), 1307(m), 1262(m), 1218(m), 1179(m), 1090(s), 994(s), 815(s), 772(m), 613(m), 578(m), 546(m), 503(m). ^1^H NMR (CDCl_3_, δ), 8.10 (d, *J*(H,H) = 8.9 Hz, 2H), 7.99 (d, *J*(H,H) = 8.6 Hz, 2H), 7.51 (d, *J*(H,H) = 8.6 Hz, 2H), 7.09 (d, *J*(H,H) = 8.4 Hz, 2H), 6.97 (d, *J*(H,H) = 8.9 Hz, 2H), 6.68 (d, *J*(H,H) = 8.4 Hz, 2H), 4.61 (s, 2H), 3.91 (s, 3H), 2.59 (q, *J*(H,H) = 7.6 Hz, 2H), 1.23 (t, *J*(H,H) = 7.6 Hz, 3H) ppm. ^13^C NMR (CDCl_3_, δ), 194.2, 171.5, 164.0, 145.0, 140.3, 133.9, 133.3, 132.4, 129.3, 129.2, 129.0, 128.8, 121.7, 113.7, 113.2, 55.5, 50.8, 28.0, 16.0 ppm. Accurate mass measurement [CI^+^, *m*/*z*]: found 408.1366 [M + H]^+^, calculated mass for C_24_H_22_ClNO_3_H: 408.1367.

*N*-(4-Bromophenyl)-*N*-(2-(4-chlorophenyl)-2-oxoethyl)-4-methoxybenzamide (**6**). Gray solid (83% yield). M.p. 150–152 °C. Selected IR (KBr, cm^−1^), 1680(s), 1601(s), 1574(m), 1513(m), 1487(m), 1427(s), 1301(s), 1260(s), 1166(s), 1025(m), 926(m), 844(s), 816(m), 772(s), 696(m), 613(s), 547(s), 503(m), 484(m). ^1^H NMR (CDCl_3_, δ), 8.09 (d, *J*(H,H) = 8.9 Hz, 2H), 7.98 (d, *J*(H,H) = 8.6 Hz, 2H), 7.52 (d, *J*(H,H) = 8.6 Hz, 2H), 7.32 (d, *J*(H,H) = 8.9 Hz, 2H), 6.97 (d, *J*(H,H) = 8.8 Hz, 2H), 6.61 (d, *J*(H,H) = 8.8 Hz, 2H), 4.57 (s, 2H), 3.91 (s, 3H) ppm. ^13^C NMR (CDCl_3_, δ), 193.5, 171.3, 164.0, 145.8, 140.6, 133.0, 132.4, 132.1, 129.3, 129.2, 121.6, 114.7, 113.8, 110.8, 109.7, 55.5, 50.2 ppm. Accurate mass measurement [CI^+^, *m*/*z*]: found 458.0157 [M + H]^+^, calculated mass for C_22_H_17_BrClNO_3_H: 458.0159.

*N*-(2-(4-Bromophenyl)-2-oxoethyl)-*N*-(4-ethylphenyl)cinnamamide (**7**). Yellow solid (79% yield). M.p. 126–127 °C. Selected IR (KBr, cm^−1^), 1697(s), 1656(vs), 1620(s), 1584(s), 1510(s), 1401(m), 1376(s), 1327(m), 1209(s), 1069(s), 1005(s), 982(s), 840(m), 812(s), 703(s), 698(s), 569(m), 549(s). ^1^H NMR (CDCl_3_, δ), 7.87 (d, *J*(H,H) = 7.6 Hz, 2H), 7.73 (d, *J*(H,H) = 15.6 Hz, 1H), 7.63 (d, *J*(H,H) = 7.8 Hz, 2H), 7.38–7.26 (m, 8H), 6.49 (d, *J*(H,H) = 15.6 Hz, 1H), 5.18 (s, 2H), 2.72 (q, *J*(H,H) = 7.6 Hz, 2H), 1.29 (t, *J*(H,H) = 7.6 Hz, 3H) ppm. ^13^C NMR (CDCl_3_, δ), 192.9, 166.4, 144.3, 142.7, 140.0, 135.1, 134.1, 132.1, 129.9, 129.6, 129.5, 129.1, 128.7, 128.1, 128.0, 118.0, 56.5, 28.5, 15.4 ppm. Accurate mass measurement [CI^+^, *m*/*z*]: found 448.0916 [M + H]^+^, calculated mass for C_25_H_22_BrNO_2_H: 448.0912.

*N*-(2-(4-Chlorophenyl)-2-oxoethyl)-*N*-(4-ethylphenyl)cinnamamide (**8**). Dark yellow solid (80% yield). M.p. 155–157 °C. Selected IR (KBr, cm^−1^), 1697(s), 1653(s), 1617(s), 1587(s), 1510(s), 1402(m), 1380(s), 1329(m), 1214(s), 1089(s), 1000(m), 981(s), 819(s), 766(m), 703(m), 548(m), 525(m). ^1^H NMR (CDCl_3_, δ), 7.92 (d, *J*(H,H) = 7.6 Hz, 2H), 7.70 (d, *J*(H,H) = 15.1 Hz, 1H), 7.43 (d, *J*(H,H) = 7.8 Hz, 2H), 7.28–7.21 (m, 8H), 6.45 (d, *J*(H,H) = 16.1 Hz, 1H), 5.16 (s, 2H), 2.68 (q, *J*(H,H) = 7.6 Hz, 2H), 1.25 (t, *J*(H,H) = 7.6 Hz, 3H) ppm. ^13^C NMR (CDCl_3_, δ), 192.8, 166.5, 144.4, 142.8, 140.1, 140.0, 135.2, 133.7, 129.7, 129.5, 129.1, 129.0, 128.8, 128.1, 128.0, 118.0, 56.6, 28.6, 15.5 ppm. Accurate mass measurement [CI^+^, *m*/*z*]: found 404.1408 [M + H]^+^, calculated mass for C_25_H_22_ClNO_2_H: 404.1412.

*N*-(4-Bromophenyl)-*N*-(2-(4-chlorophenyl)-2-oxoethyl)cinnamamide (**9**). Off-white solid (76% yield). M.p. 174–176 °C. Selected IR (KBr, cm^−1^), 1696(s), 1657(s), 1617(s), 1587(m), 1486(vs), 1408(m), 1373(s), 1207(s), 1092(m), 1070(m), 1013(s), 816(s), 763(m), 696(m), 484(s). ^1^H NMR (CDCl_3_, δ), 7.95 (d, *J*(H,H) = 8.6 Hz, 2H), 7.74 (d, *J*(H,H) = 15.5 Hz, 1H), 7.58 (d, *J*(H,H) = 8.6 Hz, 2H), 7.48 (d, *J*(H,H) = 8.6 Hz, 2H), 7.39–7.32 (m, 6H), 6.46 (d, *J*(H,H) = 15.5 Hz, 2H), 5.18 (s, 2H) ppm. ^13^C NMR (CDCl_3_, δ), 192.4, 166.1, 143.5, 141.5, 140.2, 134.8, 133.4, 132.9, 130.0, 129.9, 129.5, 129.2, 128.8, 128.1, 122.0, 117.3, 56.3 ppm. Accurate mass measurement [CI^+^, *m*/*z*]: found 456.0184 [M + H]^+^, calculated mass for C_23_H_17_BrClNO_2_H: 456.0189.

#### 3.2.3. General Procedure for Synthesis of Compounds 10–12

A mixture of the appropriate benzamide or cinnamide with an equivalent of **WR** in dry toluene was refluxed for 6 h. Following cooling to room temperature and filtration to remove unreacted solid, the filtrate was evaporated in vacuo, the residue was dissolved in 2 mL of dichloromethane and purified by silica gel column chromatography (1:1 hexane/dichloromethane as eluant) to give the products **10**–**12**. Cinnamides **7**–**9** did not show any reaction with **WR**, returning the starting materials.

*N*-(4-Bromophenethyl)-*N*-(4-ethylphenyl)-4-methoxybenzoselenoamide (**10**). Greyish yellow paste (0.25g, 50%). Selected IR (KBr, cm^−1^), 1601(s), 1508(s), 1488(m), 1448(s), 1399(s), 1300(m), 1250(vs), 1170(s), 1073(m), 1029(m), 830(m), 806(m). ^1^H NMR (CD_2_Cl_2_, δ), 8.03 (d, *J*(H,H) = 8.0 Hz, 1H), 7.45 (d, *J*(H,H) = 8.4 Hz, 2H), 7.26–7.23 (m, 4H), 7.10–7.02 (m, 3H), 6.94 (d, *J*(H,H) = 8.0 Hz, 2H), 6.10 (d, *J*(H,H) = 8.4 Hz, 2H), 4.72 (t, *J*(H,H) = 8.1 Hz, 2H), 3.74(s, 3H), 3.24 (t, *J*(H,H) = 8.1 Hz, 2H), 2.60 (q, *J*(H,H) = 7.6 Hz, 2H), 1.21 (t, *J*(H,H) = 7.6 Hz, 3H) ppm. ^13^C NMR (CD_2_Cl_2_, δ), 206.6, 159.5, 143.7, 140.0, 137.6, 131.5, 130.7, 129.0, 128.6, 126.2, 120.2, 114.4, 112.4, 62.4, 55.2, 31.4, 28.3, 15.0 ppm. ^77^Se NMR (CDCl_3_, δ), 598.4 ppm. Accurate mass measurement [CI^+^, *m*/*z*]: found 502.0283 [M + H]^+^, calculated mass for C_24_H_24_BrNOSeH: 502.0285.

*N-*(4-Chlorophenethyl)-*N*-(4-ethylphenyl)-4-methoxybenzoselenoamide (**11**). Reddish yellow paste (0.21 g, 46%). Selected IR (KBr, cm^−1^), 1602(s), 1508(s), 1488(m), 1448(s), 1398(s), 1300(m), 1251(vs), 1170(s), 1030(m), 832(m), 809(m). ^1^H NMR (CDCl_3_, δ), 7.25–7.23 (m, 4H), 7.19 (d, *J*(H,H) = 8.6 Hz, 2H), 7.02 (d, *J*(H,H) = 8.0 Hz, 2H), 6.85 (d, *J*(H,H) = 8.0 Hz, 2H), 6.55 (d, *J*(H,H) = 8.0 Hz, 2H), 6.55 (d, *J*(H,H) = 8.0 Hz, 2H), 4.67 (d, *J*(H,H) = 8.0 Hz, 2H), 3.69 (s, 3H), 3.21 (d, *J*(H,H) = 8.0 Hz, 2H), 2.54 (q, *J*(H,H) = 8.1 Hz, 2H), 1.18 (t, *J*(H,H) = 8.0 Hz, 3H) ppm. ^13^C NMR (CDCl_3_, δ), 206.8, 159.7, 143.7, 139.8, 136.8, 132.5, 130.4, 129.2, 128.8, 128.7, 126.2, 112.7, 62.8, 55.3, 31.5, 28.4, 15.2 ppm. ^77^Se NMR (CDCl_3_, δ), 601.5 ppm. Accurate mass measurement [CI^+^, *m*/*z*]: found 458.0789 [M + H]^+^, calculated mass for C_24_H_24_ClNOSeH: 458.0790.

*N*-(4-Bromophenyl)-*N*-(4-chlorophenethyl)-4-methoxybenzoselenoamide (**12**). Pale orange paste (0.20 g, 40%). Selected IR (KBr, cm^−1^), 1601(s), 1489(m), 1485(s), 1446(m), 1392(m), 1302(m), 1251(vs), 1170(vs), 1068(m), 1011(m), 832(m), 801(m). ^1^H NMR (CDCl_3_, δ), 7.33 (d, *J*(H,H) = 8.2 Hz, 2H), 7.27–7.16 (m, 6H), 6.82 (d, *J*(H,H) = 8.2 Hz, 2H), 6.58 (d, *J*(H,H) = 8.3 Hz, 2H), 4.63 (q, *J*(H,H) = 8.4 Hz, 2H), 3.72 (s, 3H), 3.16 (d, *J*(H,H) = 8.4 Hz, 3H) ppm. ^13^C NMR (CDCl_3_, δ), 204.8, 160.0, 145.0, 139.6, 136.5, 132.7, 130.3, 129.6, 129.2, 128.8, 128.0, 121.1, 113.0, 62.3, 55.4, 29.8 ppm. ^77^Se NMR (CDCl_3_, δ), 601.4 ppm. Accurate mass measurement [CI^+^, *m*/*z*]: found 507.9580 [M + H]^+^, calculated mass for C_22_H_19_BrClNOSeH: 507.9582.

## 4. Conclusions

In summary, we have disclosed Woollins’ reagent used as a highly efficient chemoselective selenation/reduction reagent for benzamide leading to *N*-aryl-*N*-(arylenethyl)benzoselenoamides. The reported results enhance the application of Woollins′ reagent further, providing an efficient route to the preparation of the unusual substituted selenoamides.

## Data Availability

Not applicable.

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
