# Peer review of "Synthesis and Single Crystal Structures of N-Substituted Benzamides and Their Chemoselective Selenation/Reduction Derivatives"

_molecules, 2021, doi:10.3390/molecules26082367_

Round 1

Reviewer 1 Report

In the manuscript entitled “Synthesis and Single Crystal Structures of N-Substituted Benzamides and Their Chemoselective Selenation/Reduction Derivatives” the authors present the study focusing on the opportunities of Woolin’s reagent as a selenation/reduction agent. The topic of this study is attractive and up-to-date, fits well within the scope of the journal and will certainly be of interest to the readers of the journal. The manuscript is well written, while the conclusions are clearly supported by the experimental data. I warmly recommend the publication of the manuscript after minor correction of the text. Namely, in Scheme 1, the structures 7–9 are incorrectly labelled as 10–12. I found a similar mistake in Scheme 2 (labelled as 7–9 and should be 10–12) and Scheme 3 (labelled as 10–12, it should be 7–9).

Author Response

The schemes have been corrected

Reviewer 2 Report

Prof. Woollins and coworkers reported a general method for the synthesis of benzoselenoamides. Their method is very practical and potential very useful. The manuscript should be accepted after addressing the following concern:

in Scheme 3, compounds 7-9 are untouched under their standard selenation reaction condition,  can you explain a little bit more why this reaction do not work, as we compared with compounds in scheme 2, those compounds are very similar, and how did the double bond presented in 7-9 inhibit the reaction even the reduction of a carbonyl group? Additionally, did the author try other conditions for this transformation?

Author Response

We have added a comment on why we think the reaction did not proceed.

Reviewer 3 Report

This manuscript (ID:  molecules - 1170681) describes the use of 2,4-bis(phenyl)-1,3-diselenadiphosphetane-2,4-diselenide (Woollins' Reagent, WR) as a selenation/reduction reagent for the conversion of N-aryl-N-(2-oxo-2-arylethyl)benzamides to N-aryl-N-(2-arylethyl)benzoselenoamides.  This manuscript is short on results.  There are only three examples of the featured selenation/reduction reported in the manuscript.  It would be nice if the authors had shown more examples as it could potentially be useful.  A major portion of the manuscript is the five new crystal structures obtained for two of the substrates (one of which did not react with WR) and three of the products.  Perhaps the manuscript would be more appropriate for an X-ray crystallographic journal.  In its current form, it does not have enough new chemistry to interest many readers of Molecules.  I should also point out that the manuscript needs careful review of some of the English (use of plurals and articles), spelling (e.g. Michael adduct, p 2, line 71) and some style issues [alignment of rows in the Table, spacing between numbers and text, degree symbols (˚) after all angles on p 5, boldface compound numbers in 3.2.1., 3.2.2. and 3.2.4. (3.2.3. is missing), and I would include experimental details of the X-ray data collection in the Experimental rather than in the General section].  

Author Response

We submitted this manuscript for the special issue celebrrating crystallography

I have checked the grammar

We have corrected the formatting issues

Round 2

Reviewer 3 Report

I am not an X-ray crystallographer, and I did not know that Molecules had an issue devoted to crystallography.

I do not think that this manuscript presents enough work to justify publication based on a new selenation/reduction procedure.  I also do not think that the X-ray structures are particularly novel.  However, an X-ray crystallographer might think otherwise on this last point.

Line 154 should read:   ...proceeded through a Michael reaction.

Overall the manuscript is reasonably well written.